# EfficientCellSeg: Efficient Volumetric Cell Segmentation Using Context Aware Pseudocoloring

**Royden Wagner**                                    ROYDEN.WAGNER@BIOQUANT.UNI-HEIDELBERG.DE
**Karl Rohr**                                                        K.ROHR@UNI-HEIDELBERG.DE
*Biomedical Computer Vision Group, BioQuant, IPMB, Heidelberg University*

## Abstract

Volumetric cell segmentation in fluorescence microscopy images is important to study a wide variety of cellular processes. Applications range from the analysis of cancer cells to behavioral studies of cells in the embryonic stage. Like in other computer vision fields, most recent methods use either large convolutional neural networks (CNNs) or vision transformer models (ViTs). Since the number of available 3D microscopy images is typically limited in applications, we take a different approach and introduce a small CNN for volumetric cell segmentation. Compared to previous CNN models for cell segmentation, our model is efficient and has an asymmetric encoder-decoder structure with very few parameters in the decoder. Training efficiency is further improved via transfer learning. In addition, we introduce Context Aware Pseudocoloring to exploit spatial context in z-direction of 3D images while performing volumetric cell segmentation slice-wise. We evaluated our method using different 3D datasets from the Cell Segmentation Benchmark of the Cell Tracking Challenge. Our segmentation method achieves top-ranking results, while our CNN model has an up to 25x lower number of parameters than other top-ranking methods. Code and pretrained models are available at: https://github.com/roydenwa/efficient-cell-seg

**Keywords:** Volumetric cell segmentation, deep learning, pseudocoloring, context awareness, parameter efficiency.

## 1. Introduction

Cell segmentation in volumetric microscopy images is essential to study a wide variety of cellular processes. Examples of applications are the analysis of cancer cells or behavioral studies of cells in the embryonic state. In recent years, deep learning methods are frequently used for cell segmentation. The majority of current deep learning architectures for cell segmentation are convolutional neural networks (CNNs) (e.g., Arbelle and Raviv 2019; Scherr et al. 2020; Pena et al. 2020), but like in other computer vision fields, vision transformer (ViT) architectures are also used (e.g., Prangemeier et al. 2020).

Since U-Net (Ronneberger et al., 2015) was published, encoder-decoder CNNs with skip connections have become the most used CNN architecture for cell segmentation. These encoder-decoder CNNs typically perform semantic segmentation with three labels: One for background, one for cells, and one for cell borders. Chen et al. 2016 combine multiple U-Nets with a long short-term model (LSTM) to perform volumetric cell segmentation. The U-Nets act as feature extractors, and feature maps of adjacent 2D slices are processed by the LSTM part to extract hierarchical features from the 3D context. Khan et al. 2020 use a

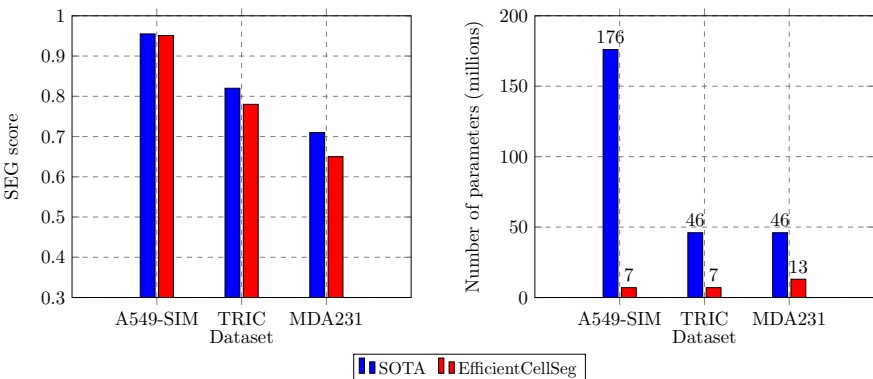

Figure 1: Parameter efficiency. Our EfficientCellSeg model achieves competitive SEG scores compared to state-of-the-art (SOTA) models (left) despite having a much lower number of parameters (right).

3D variant of the U-Net architecture to perform volumetric cell segmentation. Scherr et al. 2020 use a Dual U-Net (DU-Net) with two distinct decoder paths to perform volumetric cell segmentation slice-wise. One decoder predicts a heatmap for the distance of cells to background regions, the other decoder predicts a heatmap for the distance to neighbor cells.

A general trend is that deep learning models for cell segmentation are becoming larger, while the amount of available training data in applications remains limited. Since training large models with a large number of parameters requires a large amount of data, methods have been developed to increase the size of the training data or to adapt the training process. Recent approaches use generative adversarial networks (GANs) to extend the size of the training data by data augmentation beyond geometric transformations (Bailo et al., 2019; Naghizadeh et al., 2021) or use semi-supervised learning to exploit sparsely annotated datasets (Shailja et al., 2021; Takaya et al., 2021).

Instead of using a large CNN, we take a different approach and introduce a small encoder-decoder CNN. We propose EfficientCellSeg, a neural network model that has a much lower number of parameters than existing models for volumetric cell segmentation and thus can be trained with less training data. Our method analyzes 3D fluorescence microscopy images slice-wise using stacks of 2D slices. In previous work, 3D images are often analyzed by independently considering 2D slices, which leads to a loss of information about the spatial context in z-direction. Related methods use multiple feature extractor CNNs to process adjacent 2D slices in parallel and combine the extracted feature maps at a later stage of the methods (Kitrungrotsakul et al., 2019; Chen et al., 2016). In this way, spatial context in z-direction is used for slice-wise segmentation, but the complexity of the models is also increased significantly. For our method, we introduce Context Aware Pseudocoloring as pre-processing method to exploit spatial context of 3D images in z-direction while performing volumetric cell segmentation slice-wise. Cell segmentation results from adjacent slices are employed for pseudocoloring to provide relevant spatial context for volumetric cell segmentation. The contributions of our paper are twofold:

- We use recent advances in neural architecture search to design a CNN with an asymmetric encoder-decoder structure and only 6.7 million parameters.

- We introduce Context Aware Pseudocoloring to exploit spatial context while performing volumetric cell segmentation slice-wise.

## 2. Method

### 2.1. Model Architecture

Recent advances in neural architecture search focus on optimizing the efficiency of CNN models. Tan and Le 2019 use a compound coefficient that scales network hyperparameters such as depth, width, and resolution to obtain a new set of efficient CNNs called EfficientNets. These models achieve a higher accuracy on the ImageNet dataset while containing much less parameters than previous models. We build upon these achievements and use parts of an EfficientNet-B5 model as encoder in our EfficientCellSeg model. The main building block of EfficientNets are mobile inverted bottleneck (MBConv) blocks. EfficientNet models consist of seven high-level blocks with an increasing number of MBConv blocks per high-level block. We employ five of these high-level blocks plus the initial convolutional layer of the sixth block as encoder. EfficientNets are typically optimized for fixed input sizes. We use a fixed input size of $384 \times 384 \times 3$ voxels for images with three channels. In the encoder, the input images are downsampled to a feature tensor with a shape of $24 \times 24 \times 1056$ voxels. In the decoder, the feature tensor is upsampled with four upsampling blocks. Each of these blocks contains two convolutional layers, two batch normalization layers, and one bilinear upsampling layer. Width and height are doubled at each upsampling block, the number of output filters per convolutional layer is decreased at each block from 64 to 16, and the encoder and decoder are additionally connected with four skip connections. Our EfficientCellSeg model has an asymmetric architecture compared to existing encoder-decoder CNNs for cell segmentation (e.g., Ronneberger et al. 2015; Khan et al. 2020; Scherr et al. 2020) where encoder and decoder have a very similar structure and shape. Figure 2 shows the overall architecture of the proposed EfficientCellSeg network.

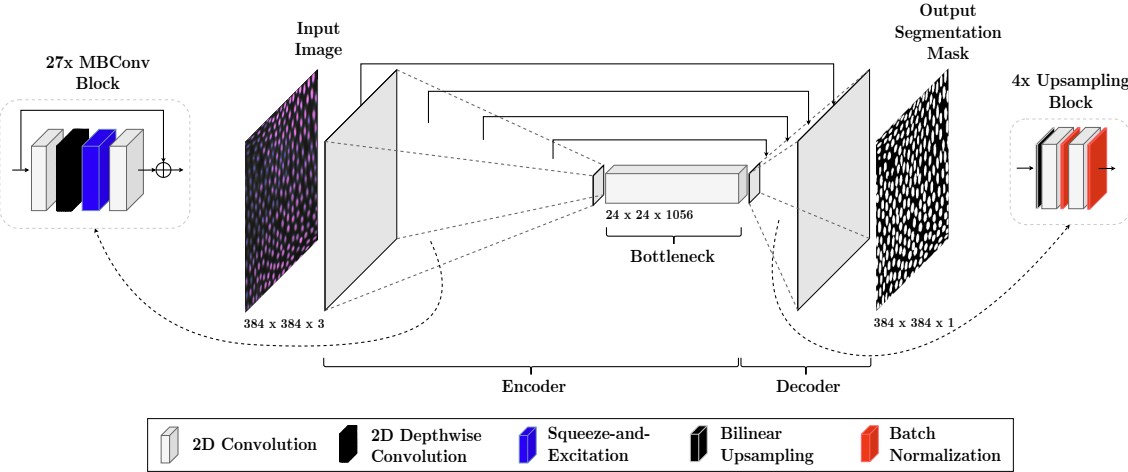

Figure 2: EfficientCellSeg architecture

| Model | Parameters overall | Parameters encoder | Parameters decoder | Output filters decoder |
|---|---|---|---|---|
| U-Net (2015) | 31.4 M | 18.9 M | 12.5 M | [512, 256, 128, 64] |
| DU-Net (2020) | 46.4 M | 22.2 M | 24.2 M | [512, 256, 128, 64] |
| EfficientCellSeg | 6.7 M | 5.6 M | 1.1 M | [64, 48, 32, 16] |

Table 1: Model sizes. We count the parameters in the bottleneck parts of all models as part of the encoder. The column "Output filters decoder" gives the number of output filters in the convolutional layers per block in the decoder part.

Table 1 provides a comparison of the model sizes of a standard U-Net (Ronneberger et al., 2015), a more recent Dual U-Net (DU-Net) model (Scherr et al., 2020), and our EfficientCellSeg model. We considered publicly available implementations of the U-Net[1] and DU-Net[2] models as reference. Since both the U-Net and DU-Net are fully convolutional models with a fixed number of output filters per convolutional layer, the number of parameters is invariant to different input shapes. In Table 1 it can be seen that for our EfficientCellSeg model, the number of overall parameters and the number of parameters in the decoder are much lower compared to the other models. EfficientCellSeg has roughly five times the number of parameters in the encoder compared to the decoder, whereas other encoder-decoder CNNs such as U-Net or DU-Net have roughly the same number of parameters in their encoder and decoder parts.

The underlying idea for reducing the number of parameters in the decoder of our model is as follows. The encoder acts as feature extractor that determines visual features with an increasing complexity through its layers. The decoder maps these features to intermediate representations and ultimately to pixels in the input image to perform pixels-wise predictions. Compared to other recent methods (Scherr et al., 2020; Pena et al., 2020), we simplify the classification task by only distinguishing between two classes, background and cells, instead of three or more classes. We assume that a smaller number of mapping combinations of the extracted features is sufficient to solve the classification task which reduces the number of parameters in the decoder part of our model.

### 2.2. Context Aware Pseudocoloring

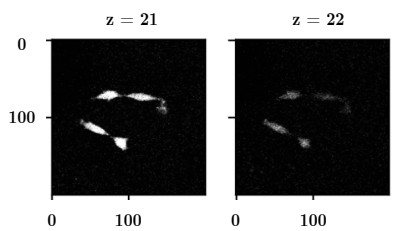

Figure 3: Decreasing visibility of cells in adjacent 2D slices.

The proposed cell segmentation method analyzes 3D images slice-wise using stacks of 2D slices. The spatial context of a 2D slice are the adjacent 2D slices. Many cell types have an ellipsoidal shape, thus the visibility decreases in z-direction towards the cell borders (see Figure 3). For theses slices with low-contrast cell parts, information from adjacent slices can be exploited to improve the segmentation result in the current slice.

In our method, we use Context Aware Pseudocoloring to exploit information from adjacent slices (the previous slice $z - 1$ and the next slice $z + 1$) for

1. https://github.com/zhixuhao/unet
2. https://git.scc.kit.edu/KIT-Sch-GE/2021_segmentation

the current slice. We generate three pseudocolor channels. First, the two adjacent slices are processed with contrast limited adaptive histogram equalisation (CLAHE) filters (Pizer et al., 1987). Then, regions of interest (ROIs) are determined in the two adjacent slices via thresholding. The intermediate result is a rough segmentation, in most cases there is an oversegmentation. Afterwards, we perform a multiply-accumulate step, where the binary values in the segmentation mask are multiplied with the intensity values of the current slice and the result of this multiplication is added to the intensity values of the current slice (which makes the result less prone to errors in determining the ROIs). This highlights regions in the current slice where cells are located in adjacent slices. To generate the red pseuodocolor channel, we perform this procedure for the previous slice $z - 1$. The blue pseudocolor channel is generated analogously using the next slice $z + 1$. For the green pseudocolor channel the intensity values of the current slice $z$ are used. All three pseudo-color channels are combined to a pseudocolor image. Finally, the intensity values of the pseudocolor image are normalized between zero and one. Figure 4 shows the whole process.

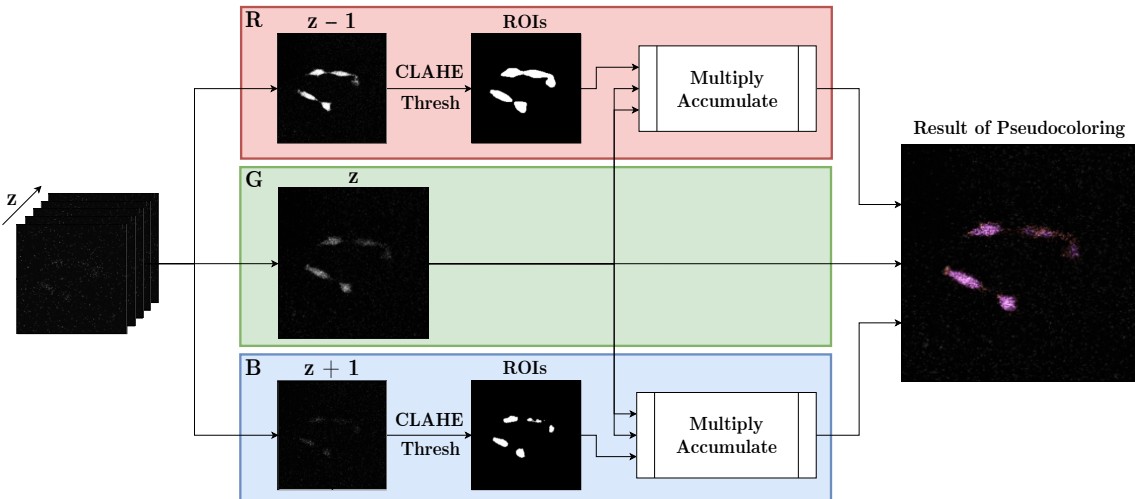

Figure 4: Context Aware Pseudocoloring

An advantage of using a pseudocolor image with three channels as input for our EfficientCellSeg model is that it simplifies the use of transfer learning. Accordingly, we can initialize the EfficientNet model in the encoder with ImageNet (Deng et al., 2009) weights (obtained from training with natural color images) without further adjustments.

## 3. Experiments

### 3.1. Datasets

We evaluated our method using three different 3D fluorescence microscopy datasets from the Cell Segmentation Benchmark of the Cell Tracking Challenge (Ulman et al., 2017). The Fluo-C3DL-MDA231 (MDA231) dataset consists of 3D images of human breast carcinoma cells with a size of $512 \times 512 \times 30$ voxels. The Fluo-N3DL-TRIC (TRIC) datset comprises 3D images of a developing Drosophila Melanogaster embryo with a size of $1745 \times 2440 \times 13$

voxels. The Fluo-C3DH-A549-SIM (A549-SIM) dataset consists of 3D images of lung cancer cells with a size of $350 \times 300 \times 29$ voxels or $400 \times 300 \times 37$ voxels. We originally developed the Context Aware Pseudocoloring method for the MDA231 dataset. In this dataset, many cases of cells with strongly decreasing visibility in adjacent slices exist, as in Figure 3. The A549-SIM dataset contains only one cell per 3D image, but also there the visibility decreases at cell borders. The same applies to the large number of cells in the TRIC dataset.

### 3.2. Impact of Transfer Learning, Context Aware Pseudocoloring, and Decoder Size

To investigate the impact of transfer learning and Context Aware Pseudocoloring of our method, we trained three EfficientCellSeg models with different configurations using the A549-SIM training dataset. For all three models, we used Adam (Kingma and Ba, 2015) as optimizer, a Dice loss function (Milletari et al., 2016), 1488 training samples, 372 test samples, and trained for 100 epochs. In all cases, the decoder was initialized with random weights. The encoder of Model 1 was initialized with random weights and we generated 2D training samples with three channels by stacking three identical copies of each 2D slice. The encoder of Model 2 was initialized with ImageNet weights and we generated 2D training samples as for Model 1. The encoder of Model 3 was initialized with ImageNet weights and the model was trained with pseudocolored 2D slices. Figure 5 shows the training curves.

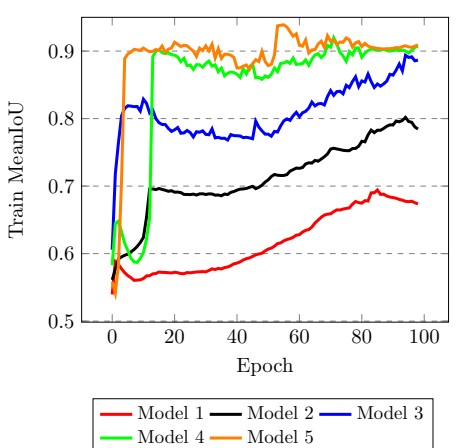

Figure 5: Impact of transfer learning, Context Aware Pseudocoloring, and decoder size.

The performance is quantified by the mean intersection over union (MeanIoU), which measures the similarity of the segmentation result and the reference annotations:

$$MeanIoU = \frac{1}{C} \sum_C \frac{TP_C}{TP_C + FP_C + FN_C}$$

For the two class labels (C), background and cells, the true positive (TP), false positive (FP), and false negative (FN) pixels are determined.

Models 2 and 3 (initialized with ImageNet weights) reach a higher MeanIoU value than Model 1. Furthermore, a significant improvement is achieved by using Context Aware Pseudocoloring (Model 3 compared to Model 2). This shows that exploiting spatial context in adjacent slices in this way improves the result and works well in combination with ImageNet weights.

We also investigated the impact of the number of parameters in the decoder using the A549-SIM training dataset. We trained two more models with the same setup as Model 3 but with an increased number of parameters in the decoder ($4.8\,\mathrm{M}$ and $12.5\,\mathrm{M}$ vs. $1.1\,\mathrm{M}$). Figure 5 shows that Models 4 and 5 converge faster than Model 3 in this experiment and yield somewhat higher MeanIoU scores for the *training* samples. However, for the *test* samples, the MeanIoU scores of Models 3, 4, and 5 are very similar (see Table 2). Therefore, it can be concluded that more parameters in the decoder do not improve the segmentation

performance in this experiment. Corresponding to the lower number of parameters, Model 3 requires less time for training than Models 4 and 5 for one epoch ($T_{Epoch}$) on a Nvidia® Tesla® P100 GPU, and the inference times per sample are faster on the used GPU ($TI_{GPU}$) and an Intel® Xeon® CPU ($TI_{CPU}$).

| Model | Output filters decoder | Parameters decoder $\Downarrow$ | Test MeanIoU $\Uparrow$ | $T_{Epoch}$ $\Downarrow$ | $TI_{GPU}$ $\Downarrow$ | $TI_{CPU}$ $\Downarrow$ |
|---|---|---|---|---|---|---|
| 3 (—) | [64, 48, 32, 16] | 1.1 M | 0.9007 | 111 ms | 79.5 ms | 334 ms |
| 4 (—) | [112, 80, 48, 16] | 4.8 M | 0.9003 | 127 ms | 80.3 ms | 652 ms |
| 5 (—) | [512, 256, 128, 64] | 12.5 M | 0.9042 | 170 ms | 92.0 ms | 1.47 s |

Table 2: Impact of decoder size.

### 3.3. Cell Segmentation Benchmark

To evaluate the performance of our method, we trained the EfficientCellSeg models with the available training datasets for each of the three used datasets from the Cell Segmentation Benchmark of the Cell Tracking Challenge and participated in the challenge[3].

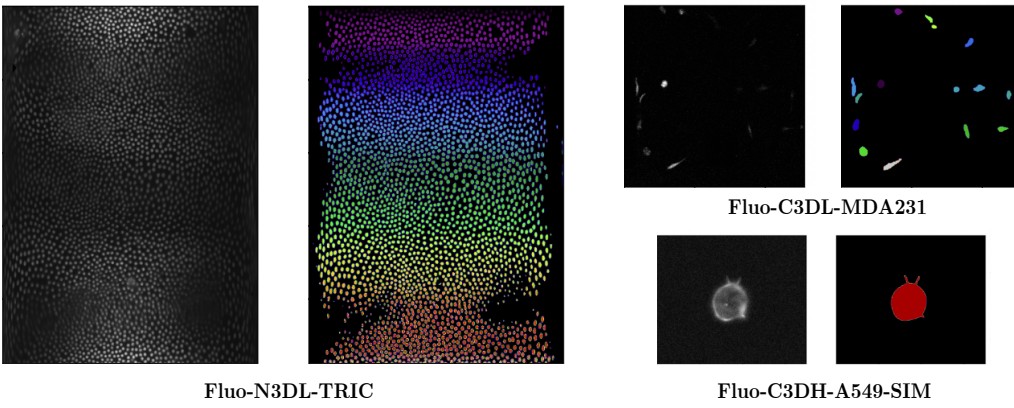

**Fluo-C3DL-MDA231**

**Fluo-N3DL-TRIC**

**Fluo-C3DH-A549-SIM**

Figure 6: Example 2D slices of 3D images from the datasets of the Cell Segmentation Benchmark and corresponding segmentation results of EfficientCellSeg.

For the MDA231 and A549-SIM datasets, we resized the 2D slices to $384 \times 384$ pixels. Since the TRIC dataset has much larger image sizes, we applied a sliding window scheme with six square patches and resized these patches to $384 \times 384$ pixels. We used Context Aware Pseudocoloring for all datasets and initialized all models with ImageNet weights. During training, we used Adam as optimizer and reduced the learning rate at plateaus. We trained separate models for the three considered datasets. The MDA231 dataset contains many irregularly shaped cells that are difficult to distinguish. Hence, we additionally trained a second model to predict heatmaps for the cell centers (similar as in, e.g., Schmidt et al. 2018). The segmentation model was trained using a Dice loss and the cell center heatmap model was trained using a focal loss (Lin et al., 2018). In a post-processing step, the predicted cell centers were used as markers for a watershed algorithm (Beucher and Meyer,

---

3. http://celltrackingchallenge.net/participants/HD-Wag-GE

1993) to label individual cells. Since all images in the A549-SIM dataset contain only one cell, we used only the segmentation model. The TRIC dataset contains many cells with an ellipsoidal shape. Therefore, we used the segmentation model, and as post-processing a distance-based watershed algorithm. Figure 6 shows example 2D slices of 3D images from the challenge datasets and the corresponding segmentation results of our method.

As performance metric, we used the SEG score (Maška et al., 2014), which is an object-based Jaccard index that measures how well the segmented regions (S) match ground truth annotations (R). The metric is defined as:

$$SEG(R, S) = \begin{cases} \frac{|R \cap S|}{|R \cup S|}, & \text{if } |R \cap S| > 0.5 \cdot |R|, \\ 0 & \text{else} \end{cases}$$

We additionally introduce the parameter efficiency with respect to the SEG score (PESEG) as metric, which relates the achieved SEG score to the number of model parameters N:

$$PESEG(SEG, N) = \frac{SEG}{N} \cdot 10^6$$

Table 3 shows the obtained results as well as results of state-of-the-art methods. For each dataset, we selected the best method of the challenge. Our EfficientCellSeg model achieves competitive SEG scores as the best methods despite having much less parameters. The best method for the A549-SIM dataset uses an ensemble of four 3D U-Nets with a total of 176 million parameters (DKFZ-GE)(Isensee et al., 2021). In contrast, we use only one EfficientCellSeg model, which has 25 times fewer parameters. The best method for the MDA231 and the TRIC datasets uses DU-Net models (KIT-Sch-GE)(Scherr et al., 2020). For the TRIC dataset, we use only one EfficientCellSeg model, which has a factor of 6 less parameters. For the MDA231 dataset, we have a factor of 3 less parameters, although we use an additional EfficientCellSeg to determine cell centers. Accordingly, our method achieves much higher PESEG scores for all three datasets.

| Dataset | Method | CNN model | Parameters overall ⇓ | SEG ⇑ | Ranking | PESEG ⇑ |
|---------|--------|-----------|----------------------|-------|---------|---------|
| A549-SIM | DKFZ-GE | 4 x 3D U-Net | 176.0 M | 0.955 | 1/12 | 0.005 |
|  | Ours | EfficientCellSeg | 6.7 M | 0.951 | 2/12 | 0.142 |
| TRIC | KIT-Sch-GE | DU-Net | 46.4 M | 0.821 | 1/8 | 0.018 |
|  | Ours | EfficientCellSeg | 6.7 M | 0.782 | 3/8 | 0.117 |
| MDA231 | KIT-Sch-GE | DU-Net | 46.4 M | 0.710 | 1/19 | 0.015 |
|  | Ours | 2 x EfficientCellSeg | 13.4 M | 0.646 | 2/19 | 0.048 |

Table 3: Results for different 3D datasets of the Cell Segmentation Benchmark.

## 4. Conclusion

We have presented a novel method for volumetric cell segmentation that achieves competitive results to state-of-the-art methods despite having much less parameters. Our EfficientCellSeg model comprises an efficient feature extractor as encoder and a decoder with a reduced number of output filters in the convolutional layers. Furthermore, we have introduced Context Aware Pseudocoloring to exploit spatial context of 3D images while performing volumetric cell segmentation slice-wise.

## Acknowledgments

Support of the DFG (German Research Foundation) within the SFB 1129 (project Z4) and the SPP 2202 (RO 2471/10-1) and the BMBF within de.NBI is gratefully acknowledged.

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
