# OpenReview forum: "EfficientCellSeg: Efficient Volumetric Cell Segmentation Using Context Aware Pseudocoloring"
_MIDL.io/2022/Conference — MIDL 2022_

### Official Review · Reviewer_3GAh · 2022-01-23

**Confidence:** 3
**Preliminary Rating:** 4
**Recommendation:** Poster

**Summary:**

Volumetric cell segmentation in fluorescence microscopy images is an important research topic. Due to the limited number of available 3D fluorescence microscopy images, this paper proposes a small CNN for the segmentation. The proposed CNN model uses an asymmetric encoder-decoder structure and has very few parameters in the decoder. Transfer learning is exploited for efficient training, and Context Aware Pseudocoloring is introduced to exploit the spatial context in the z-direction. The proposed method was validated on benchmark datasets and achieved top-ranking results.


**Strengths:**

- The idea of addressing the problem of limited data with a smaller network model that has a reduced number of parameters is sensible.
- The information in the z-direction is exploited so that consistency between slices can be exploited.

**Weaknesses:**

- The limitations of existing methods are not clear.
- There are no experimental results that support the rationale for simplifying the decoder.
- The performance of the proposed method was not top-1 for any of the datasets considered in the experiments.

**Deanonymize Review:**

no

**Detailed Comments:**

- Existing methods augment the training data with GANs or use semi-supervised learning. What are the limitations of these methods? Why does the proposed method choose not to use these techniques?
- The authors have explained why the decoder is simplified to reduce the number of network weights. Is there any experiment as well to support this argument? For example, what would the results be if the encoder is simplified or both of them are simplified?
- It should be indicated in Table 2 whether higher or lower values are better for each evaluation metric.
- Although the proposed method ranked top-3 for the given datasets, it was never the top-1 method. I am wondering what difference between the proposed method and the best methods could lead to the difference in performance. Some discussion on this can be given.
- I am wondering what the practical value of the PESEG score is. The motivation of reducing the number of parameters is to deal with limited data. But ultimately the goal is to achieve accurate segmentation. As long as the segmentation is accurate, do we care about how many parameters are used?

**Final Rating After The Rebuttal:**

4: Weak Accept

**Justification Of The Final Rating:**

Overall, the response is satisfactory. However, it is still not clear what the limitations of GANs or semi-supervised methods are. Why do they help only to a certain extent? To what extent? Why is the proposed strategy better? Also, considering that the performance is never top-1, my recommendation remains.

**Paper Type:**

methodological development

**Questions To Address In The Rebuttal:**

- The limitations of existing methods should be elaborated.
- The rationale for simplifying the decoder should also be supported by experimental results.
- The authors can explain why the PESEG score matters.


**Special Issue:**

no

---

### Official Review · Reviewer_Nx4J · 2022-01-23

**Confidence:** 5
**Preliminary Rating:** 4
**Recommendation:** Poster

**Summary:**

The authors present a method for 3D instance segmentation that introduces two novel approaches:

- (Pre-processing) A pseudocoloring of the 2D input image (z-slice) to gather some information from the Z dimension as a preprocessing

- (CNN architecture) An encoder-decoder with an important decrease in the number of trainable parameters and which uses a pretrained encoder.

The proposed method is evaluated with data from the cell tracking challenge. It achieves competitive accuracy results (top-3 rankings) with a reduction between 70-97% in the number of parameters when compared with 3D U-Nets and Double UNets.

**Strengths:**

Deep neural networks have shown a great potential to synthesise complex and heterogeneous information from large set of images. In computer vision it is common to find architectures with ~100M parameters. However, such architectures might not be efficient when we comparing the ration between the achieved accuracy and the number of parameters. The research perform by the authors is interesting because first, it exploits (successfully) the idea of extracting the information in an improved manner (i.e. with less parameters), and second, it contributes to the use of competitive architectures with limited hardware (GPU) resources. Additionally, they propose the use of pretrained encoders, which can reduce the time needed for training.


**Weaknesses:**

The authors claim that their method can be used for 3D cell segmentation but I think they could exploit its applicability a bit more or at least add a further discussion about it:

- The method could be used for 2D segmentation: Although the pseudocoloring preprocessing would not apply in this case, it would be interesting to know what is the performance of the proposed architecture compared to other submissions to the CTC. Indeed, it might be a more appropriate way of evaluating the performance of the CNN.
- Although 3D and 2D+t have different spatial distribution, in 2D+t fluorescence videos, photobleaching is a common limitation. The authors could consider to try in the future the proposed pseudocoloring to improve the results in those situations.










**Deanonymize Review:**

no

**Detailed Comments:**

In the abstract the authors talk about efficiency but that term can be ambiguous. I would suggest to be more explicit and clearly state that they refer to the number of trainable parameters.

Section 4, Related Work, is placed at the end of the manuscript. I would recommend the authors moving it to the introduction so the reader can understand better what are the main contributions of the proposed work.

The proposed pseudocoloring is mainly for fluorescence microscopy. I think this should be clearly stated in the text. Otherwise, I would suggest the authors to give an intuition about how would it work with electron microscopy, DIC or phase contrast microscopy images.


In Figure 4, the general color of the output is magenta while the slice z+1 is brighter than z -1. I would expect this image to be more blueish. Could the authors please confirm that it is correct?

In page 5, the authors say: "An advantage of using a pseudocolor image with three channels as input for our EffficientCellSeg is that it simplifies the use of transfer learning." I think this is not entirely correct. You could either copy&paste the grey scale image in the three channels from the RGB image or convert the grey scale image into RGB. Moreover, is this the approach used for Model 2?

In page number 7, the authors say "Hence, we additionally trained a second model to predict heatmaps for the cell centers.". Such approach has already been proposed by others such as StarDist [1] or DeepWater [2,3]. I would recommend the authors to cite any reference they may have used for this.


----
[1] Uwe Schmidt, Martin Weigert, Coleman Broaddus, and Gene Myers.
Cell Detection with Star-convex Polygons.
International Conference on Medical Image Computing and Computer-Assisted Intervention (MICCAI), Granada, Spain, September 2018.

[2] F. Lux and P. Matula, "DIC Image Segmentation of Dense Cell Populations by Combining Deep Learning and Watershed," 2019 IEEE 16th International Symposium on Biomedical Imaging (ISBI 2019), 2019, pp. 236-239, doi: 10.1109/ISBI.2019.8759594.

[3] Filip Lux, Petr Matula, Cell Segmentation by Combining Marker-Controlled Watershed and Deep Learning arXiv 2020


**Final Rating After The Rebuttal:**

4: Weak Accept

**Justification Of The Final Rating:**

I thank the authors for answering the comments and updating the manuscript. The contents are now clear and the manuscript has improved the technical description of the method. All my doubts have been answered.

**Paper Type:**

methodological development

**Questions To Address In The Rebuttal:**

When looking at the pseudocoloring approach in Figure 4, I wonder whether the network would not be able to learn such process automatically. For example, the input image of the network is supposed to be an RGB image, 3 channels. If the authors introduce the slices z-1, z, z+1 as an input to the network, would it achieve a similar performance to the approach using pseudocoloring?

Why do the authors use a fixed input size? Did they downsample the images to this size or perform a tiling strategy?


**Special Issue:**

no

---

### Official Review · Reviewer_B3Jy · 2022-01-24

**Confidence:** 3
**Preliminary Rating:** 4
**Recommendation:** Poster

**Summary:**

This paper describes and validate a method for cell segmentation in 3D image stacks. The method is based on an asymmetric down and up-sampling architecture and also uses a spatial propagation method to use spatial information between slices. It shows favorable results using less parameters than other methods.

**Strengths:**

The paper is easy to read and the method is fairly well described. The network design and architecture seems valid and the results are promising. The method has also been tested on standard datasets with good results. The paper cites relevant papers.

**Weaknesses:**

While the method is sound and promising the question is if it has enough novelty to publish. The spatial propagation part ensures spatial coherence between slices and it does that by a RGB coloring scheme. The RGB coloring is done to make it possible to use pre-trained models. This could also be done by just setting the RGB channels to the same value and then treating the image as a grey scale image. It is a little hard to judge if that would have made a difference.
I think that section 4 on related work should be moved to the introduction. A comment could then be added in the discussion part if directly comparing with these methods.


**Deanonymize Review:**

no

**Detailed Comments:**

In general it is an understandable paper with a clear motivation and well defined problem. The methods is fairly well described and the results are convincing.

**Final Rating After The Rebuttal:**

5: Strong Accept

**Justification Of The Final Rating:**

I am happy with the revised version of the paper. The authors have addressed my main concerns of the paper. It is an interesting and well performing method, that is relevant for the field. In conclusion, I believe it should be accepted.

**Paper Type:**

both

**Questions To Address In The Rebuttal:**

In general, the paper reads well but I suggest to move section 4 to the introduction instead of having it in the discussion part of the paper. Secondly, a brief note on using pre-trained networks with R=G=B could be interesting.

**Special Issue:**

no

---

### Official Review · Reviewer_pkAF · 2022-01-25

**Confidence:** 5
**Preliminary Rating:** 5
**Recommendation:** Oral

**Summary:**

The manuscript presents an efficient volumetric cell segmentation using context-aware pseudocolouring. The authors adopt an efficient CNN model that has an asymmetric encoder-decoder structure with very few parameters in the decoder. Training efficiency is further improved using transfer learning. In addition, they introduced context-aware pseudocolouring to exploit spatial context in the z-direction of 3D images while performing volumetric cell segmentation slice-wise. They evaluated their method using different 3D datasets from the Cell Segmentation Benchmark of the Cell Tracking Challenge. Their segmentation method achieves top-ranking results while their CNN model has up to 25x lower parameters than other top-ranking methods.

**Strengths:**

1. Their model is efficient.
2. Thanks to the pseudocolouring is well adapted to transfer learning. The EfficientNet model can be initialized with ImageNet weights (obtained from training with natural colour images).
3. They evaluated their method in the Cell Segmentation Benchmark of the Cell Tracking Challenge.
4. Excellent performance for a lower number of parameters.


**Weaknesses:**

1. Show results for three datasets. It would be nice to evaluate the approach on the rest.
2. Move the related work section earlier in the manuscript.
3. The code is not available.
4. No information about the reduced time needed for training and inference is given.

**Deanonymize Review:**

no

**Detailed Comments:**

See the weaknesses list above.

**Final Rating After The Rebuttal:**

5: Strong Accept

**Justification Of The Final Rating:**

The work is interesting and relevant. All my questions have been nicely addressed by the authors. There are also plans to include additional information and figures to address the other reviewers' concerns. The presentation would be of interest to MIDL'22 attendees.

**Paper Type:**

both

**Questions To Address In The Rebuttal:**

1. Show results for three datasets. It would be nice to evaluate the approach on the rest of the benchmark.
2. Move the related work section earlier in the manuscript.
3. Make the code available (github repository, google colab notebook).
4. Give information about the reduced time needed for training and inference.

**Special Issue:**

yes

---

### Meta-Review · Area_Chair_Y8u2 · 2022-02-17

**Recommendation:** Accept (Oral)
**Confidence:** 5

**Metareview:**

This paper aims to solve the task of volumetric cell segmentation. The proposed solution shares the merit of efficiency and achieving competitive performance. Based on my reading and the consensus of reviewers, a decision of accept is recommended.

---

### Decision · Program_Chairs · 2022-02-28

Accept